# Deep-sea mining discharge can disrupt midwater food webs

Michael H. Dowd [1] ✉, Victoria E. Assad[1,3], Alexus E. Cazares-Nuesser [1,3], Jeffrey C. Drazen [1], Erica Goetze [1], Angelicque E. White [1] & Brian N. Popp [2]

The Clarion-Clipperton Zone contains extensive beds of polymetallic nodules on the abyssal seafloor, with vast areas (~1.5 million km²) under license for deep-sea mining. Mining companies have proposed discharging excess waste generated during nodule extraction in the lower mesopelagic and upper bathypelagic zones, which are home to a unique faunal community including zooplankton and micronekton. Here, using compound-specific isotope analysis of amino acids, we find that natural background particles larger than 6 μm form the base of the food web, but will be diluted by the same sized, nutritionally deficient mining-associated particles. Given that 53% of zooplankton taxa are particle feeders and 60% of micronekton taxa are zooplanktivores at proposed discharge depths, there is significant potential for food-web disruption. Therefore, we show that a midwater mining plume could trigger bottom-up ecosystem impacts with potentially severe consequences for the faunal community, extending beyond zooplankton and micronekton to nekton, including large marine predators.

The Clarion-Clipperton Zone (CCZ) in the Eastern Topical Pacific is a region of great interest for proposed deep-sea mining of polymetallic nodules. The International Seabed Authority, the UN established regulatory agency[1], has granted 19 exploration licenses for polymetallic nodules covering a vast area of ~1.5 million km². Previous research has documented environmental risks of mining, including biodiversity loss in seafloor communities[2,3]. In addition to the seafloor, midwater communities may also be at risk. During the mining process, nodules are collected from the abyssal seafloor, along with seawater and sediments, and transferred through a pipe to a collection ship for separation of nodules from sediment waste. This seawater-containing sediment waste, as well as pulverized nodule particles, must be returned to the ocean. Although the release depth for this waste is currently unclear, some mining operators have proposed midwater mining discharge within the lower mesopelagic/upper bathypelagic zone[4,5]. The risks to midwater communities remain poorly understood.

A midwater mining waste plume is anticipated to negatively affect mesopelagic fauna, with impacts ranging from replacing or diluting the food of filter feeders to visual impairment of active predation and bioluminescent communication[6,7]. Pelagic food webs are complex, linking small primary producers in the upper ocean to top consumers at a range of depths. Diel vertical migration from deeper daytime depths to shallower nighttime foraging depths is common, vertically mixing the food web[8,9]. Additionally, about 5–25% of the organic matter produced in the euphotic zone is exported to deeper depths through passive sinking and active transport[10,11]. The material that reaches the mesopelagic undergoes aerobic microbial decomposition and transformation, resulting in a characteristic decrease in organic carbon and nitrogen content[12,13]. The mining plume, consisting of deep abyssal sediments from <2 μm to 63 μm in size[14], is predicted to have a high concentration of low-quality particles, along with fragmented inorganic nodules. These mining waste particles could clog the respiratory and olfactory surfaces of organisms across all trophic levels and obstruct filter-feeding structures[6,15].

Of importance to midwater food webs in the CCZ region is the presence of an oxygen minimum zone (OMZ), a region characterized by particularly hypoxic conditions (to <1 μM)[16]. Oxygen gradients influence the depth distribution of zooplankton and micronekton

¹Department of Oceanography, University of Hawaiʻi at Mānoa, Honolulu, HI, USA. ²Department of Earth Sciences, University of Hawaiʻi at Mānoa, Honolulu, HI, USA. ³These authors contributed equally: Victoria E. Assad, Alexus E. Cazares-Nuesser. ✉e-mail: mdowd3@hawaii.edu

communities[17,18], as biological adaptations to low oxygen create species-specific habitat thresholds that determine where organisms reside and feed[19,20]. In the mesopelagic, zooplankton biomass may exhibit a modest increase in the lower oxycline (LO) in comparison to the OMZ core[18,21], however, highly mobile and migratory micronekton (small fish, cephalopods, and crustacea, ~2–20 cm in size[9] often reside at these depths[17,18].

Compound-specific isotope analysis of amino acids (CSIA-AA) can identify the particle size fraction that forms the base of the food web[22,23], and it can characterize trophic structure of the community[24,25]. Understanding the contribution of different particle size fractions to the base of the food web is critical, as mining plumes will introduce particles of varying abundance, nutritional content, and quality depending on their size. This knowledge is essential for assessing the potential impacts of midwater mining plumes on the faunal community at depth.

In this study, we apply a Bayesian mixing model[26] to isotopically differentiate particle sources based on size and quantify their contribution to the base of the food web. The goal of this study is to characterize the trophic sources of the faunal food web at a proposed lower mesopelagic discharge depth to evaluate how the introduction of a midwater mining waste plume could disrupt it. We examine both the composition of mining discharge plume material and the nutritional quality at the base of the food web, along with the trophic modes of the animals living there. This work provides a critical step towards understanding the potential impacts of full-scale mining on the mesopelagic community.

## Results and discussion

During this study, we collected 3 types of particle samples: background, discharge, and plume samples in 3 size fractions, small (0.7–6 μm), medium (6–53 μm), and large (>53 μm) (Fig. 1). Background samples were collected at various depths (800–1500 m) during 3 cruises, either when no mining activity was occurring or at locations and depths far from mining activity, ensuring that these particles represent natural midwater particles. Discharge and plume samples were collected in Fall 2022 during a small-scale test mining operation. During this operation, a collector vehicle on the abyssal seafloor collected sediments and nodules, which were brought to a surface ship. The nodules were separated from abyssal sediments onboard, and the effluent waste was released at ~1250 m. This event represented a rare opportunity to study midwater discharge associated with deep-sea mining. Discharge particle samples consisted of shipboard effluent waste collected after nodules were removed but before midwater release. Plume particle samples were collected in situ after the mining waste was discharged, using transmissometry to measure turbidity and confirm that we were sampling within the plume.

Background particle data were obtained as a proxy for the potential base of the food web from previous cruises where particles, zooplankton, and micronekton were collected. Isotopic analysis was performed to characterize these particles for use in a mixing model to identify the base of the food web. This approach allows us to evaluate how the introduction of mining waste might affect the pelagic community in the future. Because plume generation lasted a maximum of 32 hours and no animal samples were collected during the Fall 2022 cruise, we lack direct data on the incorporation of plume material into the food web.

We found that mining-associated particles were nutritionally poor. Background particle amino acid concentrations were 4.7 ± 2.7, 41.1 ± 25.3, and 46.3 ± 34.7 ngN/μgPN for small, medium, and large particles, respectively. In contrast, plume and discharge particle amino acid concentrations were substantially lower at 3.8 ± 4.4, 1.7 ± 1.5, and 4.2 ± 4.7 ngN/μgPN (Fig. 1 and Table S1). There was no significant difference in amino acid concentration of small particles between

background and plume/discharge samples ($p = 0.663$, one-way ANOVA, Table S2). However, medium and large background particles had significantly higher amino acid concentrations compared to plume/discharge particles ($p_{Medium} = 0.028$, $p_{Large} = 0.035$, one-way ANOVA, Fig. 1 and Table S2). These findings indicate that the medium and large particles in the plume were significantly depleted in amino acid content relative to background material.

Background particle concentrations were much lower than plume particle concentrations. During the Fall 2022 plume sampling, we used a LISST (Laser In Situ Scattering and Transmissometry, LISST-DEEP, Sequoia Scientific) mounted on the CTD rosette to measure in situ particle concentration and size in 32 logarithmically spaced classes centered between 1.36 and 230.14 μm. As the medium and large plume and discharge particle pools were significantly different in amino acid content compared to background particles, samples were pooled into small (0.7–6 μm) and large (>6 μm, medium + large) size fractions for this analysis. Background particle concentrations were 0.08 and 0.23 μL/L for small and large size fractions, respectively, while plume particle concentrations were 9.80 and 2.18 μL/L for the same size fractions (Fig. 2 and Table S3). Background particle counts were 0.014 and $0.003 × 10^8$ particles/L for small and large size fractions,

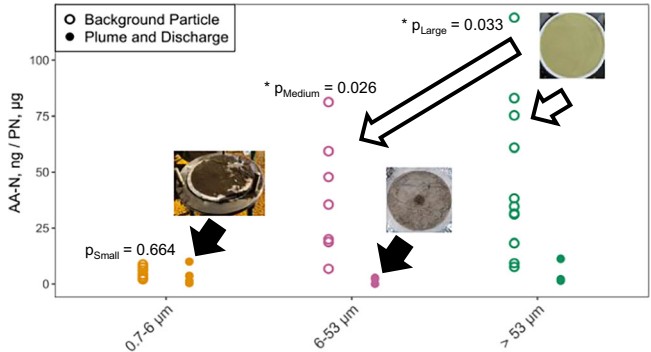

**Fig. 1 | Total amino acid concentration in particles.** Plot of the total amino acid concentration normalized to PN content, measured in 0.7–6 μm (yellow), 6–53 μm (pink), and >53 μm (green) size fractions of background particles (outliers removed) and plume particles collected from in situ filtration collected from 700 to 1250 m. Also included are discharge particles collected aboard the test mining vessel prior to mine waste being discharged at depth. P values shown are significant differences in medium and large background and plume/discharge material from one-way ANOVA (Table S2). Pictures show background particle filter (open arrow/circles) and discharge/plume filters (closed arrow/circles). Source data are provided in Source data 1.

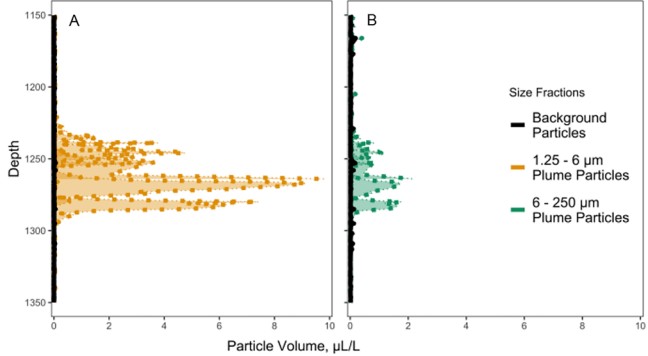

**Fig. 2 | LISST particle concentration.** Plots of particle concentration from LISST in background casts (3 casts, black) and plume casts (5 casts) in **A**) 1.25–6 μm (yellow) size fraction and **B**) 6–250 μm (green) size fraction. Data were binned to 1-db. Source data are provided in Source data 2.

respectively, compared to plume particle counts of 11.15 and $0.12 \times 10^8$ particles/L for small and large size fractions (Table S4).

Previous analyses of CCZ abyssal seafloor sediments suggest that sedimentary particles range from <2 μm to 63 μm in size[14]. The plume particle size distribution showed maxima between 2.6 and 4.3 μm (Fig. S1). Visual analysis of plume samples corroborates this pattern, showing an increase of small (0.7–6 μm) particles in the plume compared to the background. These particles were likely sedimentary in nature due to its light brown color, while dark pulverized nodule particles were observed in the 6–53 μm size range (Fig. 1). It should be noted that the "plume" is not a well-defined entity but rather an area where particle concentrations form a gradient exceeding background levels (see Supplementary Note 1). Some parts of this plume may be highly opaque, while others are only slightly turbid. Consequently, our particle concentrations represent a point along the continuum of the plume. The maximum particle concentration and count reflect the extent of our sampling but may not necessarily represent the true maximum within the plume itself.

To assess the potential impact of a plume on the faunal community, it is first necessary to understand the current contribution of particles as trophic sources to the base of the food web. Using $\delta^{15}N$ values of source and $\delta^{13}C$ values of essential amino acids (see Supplementary Note 2, Tables S5 and S6, and Fig. S2), we constructed a Bayesian mixing model to estimate plausible ranges to consumers. While diel vertical migration (DVM) is a common phenomenon and occurs at this site, an epipelagic source was not included in the mixing model, a decision based on several considerations. First, the consumer samples analyzed were collected during nighttime tows, meaning that most individuals sampled were likely non-migratory residents. Second, only ~20% of the zooplankton biomass present below the OMZ in the deep mesopelagic migrates to the upper ocean at night. Finally, the micronekton taxa included in this analysis are non-migratory, so any influence from an epipelagic DVM source from DVM should be minimal.

We found that particles >6 μm make up a significant proportion of the base of the food web in lower mesopelagic and upper bathypelagic depths below the OMZ (Fig. 3). Of the 46 animal samples collected below the OMZ (700 to 1500 m), these large particles accounted for at least 50% of the food web trophic base in 30 samples. Specifically, at the discharge depth (1000–1500 m), 16 of the 26 animal samples had large particles making up at least 50% of the food web base (Fig. 3, Source Data 3). Overall, on an individual animal basis, 65% of consumers relied on particles >6 μm for more than half of their food web base. When grouped by zooplankton size fraction/micronekton taxa, 5

of the 8 groups showed a mean contribution of particles >6 μm exceeding 50% (Fig. 3 and Table S7).

In sum, our mixing model indicates that large particles make a substantial contribution to the base of the food web below the OMZ (Fig. 3), a region expected to be impacted by mining waste plumes. Furthermore, we found that large plume and discharge particles are significantly lower in amino acid content (Fig. 1) and significantly higher in concentration compared to background particles (Fig. 2). This material would serve as the main food source for fauna, and the reduced amino acid concentration of these particles is a key indicator of their reduced nutritional value, leading to lower protein intake for the same foraging effort. Together, these findings show that a mining plume would drastically dilute the native particle supply for suspension feeders, replacing it with low nutritional quality seafloor sediments and inorganic fragmented nodules, which could inhibit the ability of organisms to meet their metabolic demands by diluting the relatively organic-rich native particles[27,28]. This alteration of the in situ particle field has serious implications for the faunal community at the depth of the proposed mining plume.

The breadth of impact of the midwater plume on pelagic food webs is likely to be high based on our knowledge of the diversity and trophic ecology of zooplankton and micronekton below the OMZ. The deep (700–1500 m) zooplankton community contained 186 amplicon sequence variants (ASVs) that were classified into 79 distinct taxa (e.g., genus, family) based on whole-community metabarcoding (18S rDNA), with 53% classified as particle feeders (proportions were nearly identical for just the discharge depth from 1000 to 1500 m; Supplementary Data 1). These taxa, including calanoid copepods, ostracods, and oncaeid copepods rely on suspended particulate organic matter as a primary food source[29–32] and may be particularly vulnerable to resuspended sediments from mining plumes. Euphausiids and midwater shrimp may also be impacted as they are known to opportunistically consume detrital aggregates at depth[33,34]. In addition, gelatinous taxa such as siphonophores, tunicates, and scyphozoans accounted for 20% of the taxa in this deep midwater community, increasing in relative read abundance with depth. These organisms may be particularly susceptible to plume-related impacts, as suspended particles can adhere to their gelatinous bodies, clog feeding structures, disrupt buoyancy, and increase mucous production and energy demands[6,7] (zooplankton community analysis detailed further in Supplementary Note 3: Functional Ecology Analysis, Zooplankton Functional Group Results). Given the importance of particle-feeding and gelatinous taxa at midwater depths, mining-induced plumes could significantly disrupt zooplankton communities and trophic interactions, with

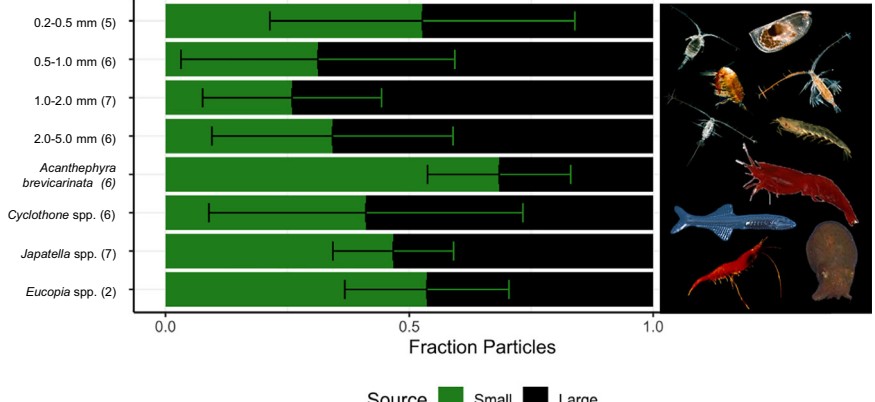

**Fig. 3 | Mixing model particle contribution to base of food web.** Plot of mixing model mean contribution of small (green; 0.7–6 μm) and large (black; >6 μm) size fractions to the trophic base of the food web for zooplankton (0.2–0.5, 0.5–1.0, 1.0–2.0, and 2.0–5.0 mm size fractions) and micronekton taxa (*A. brevicarinata*, *Cyclothone* spp., *Japatella* spp., and *Eucopia* spp.) from 700 to 1500 m. Biological replicate sample sizes are listed in parentheses. Error bars are standard deviation of mean particle contribution. *Eucopia* spp. photo © Tammy Frank. Source data are provided in Source data 3.

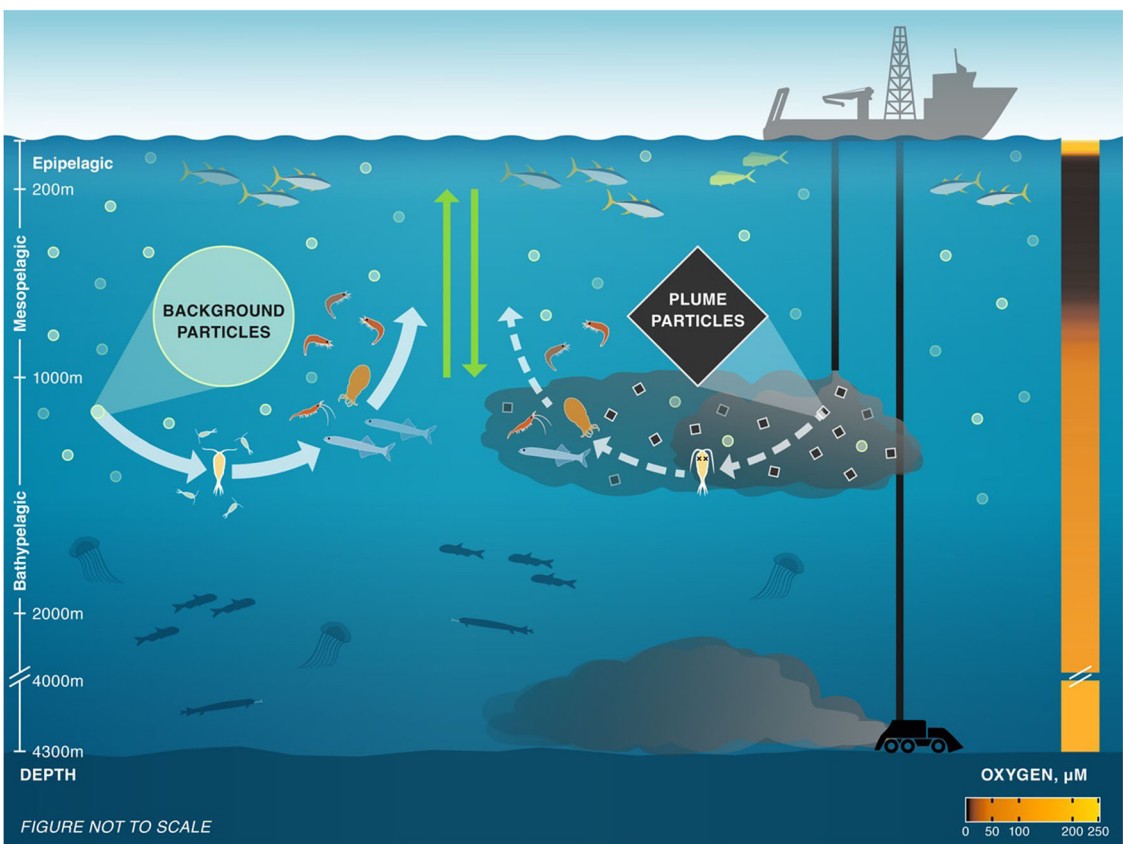

**Fig. 4 | Summary of mining waste impact on the midwater food web.** Mining-generated waste plume has the potential to disrupt the midwater food web by diluting the basal organic matter supply with low-quality organic matter-poor and inorganic particles, with impacts propagating throughout the water column. (Organisms and plume are not to scale.) Solid white arrows represent healthy food web, dashed white arrows represent disrupted mining waste-impacted food web, green vertical arrows represent vertical migration. Dissolved oxygen concentration in water column from calibrated CTD sensor. Image credit: Amanda Merrit (graphic artist). Source data are provided in Source data 4.

cascading effects on micronekton and higher trophic level predators. Indeed, we found 60% of 80 unique micronekton taxa from 700 to 1500 m were zooplanktivores, 37.5% were pelagic micronektonivores (feeding on smaller zooplanktivorous micronekton and occasionally zooplankton), 1.25% were gelatinous zooplanktivores, and 1.25% were pelagic generalists (proportions were nearly identical for just the discharge depth from 1000 to 1500 m; Supplementary Data 2). Furthermore, 85% of the density of micronekton within the discharge depth is zooplanktivores (micronekton community analysis detailed further in Supplementary Note 4). Specifically, among the micronekton sampled for CSIA-AA, *Cyclothone* spp*., Eucopia* spp., and *Japatella* spp. are zooplanktivores[35,36], whereas *Acanthephyra brevicarinata* shrimps are classified as pelagic micronektonivores[34,37]. Overall, the results suggest that the potential impacts on the zooplankton community due to mining plumes could significantly diminish the primary food source for micronekton, potentially leading to starvation or, for more mobile micronekton, migration from the area.

Accurately identifying the scale of these food web effects is challenging but essential to consider. Factors such as particle flow rate, particle size, and concentration thresholds all significantly influence the vertical and horizontal dispersion, and consequently, the volume of water affected by the sediment discharge[38,39]. Depending on the exact metrics applied, the scale of effect of the impact is estimated to be on the order of 1000 s km³/yr[38,39]. Given the potential for multiple simultaneous operations in the eastern CCZ, this represents considerable environmental risk to these communities. They provide essential ecosystem services, playing crucial roles in the biological pump[8,40], provisioning commercially harvested fishes[6,9,41], and maintaining a healthy, functioning ecosystem (Fig. 4).

There are potential options for mitigating the effects of the plume, though each comes with its own risks. Discharging at a shallower depth than the proposed discharge depth could result in similar or worse impacts, because it would affect regions of the water column with significantly higher biomass and important top predators[18,21]. Releasing the mining plume in the oxyclines above and below the OMZ core, or within the OMZ core itself, could also be especially harmful. Some zooplankton species are highly sensitive to small oxygen variations and adjust their vertical position to remain within their preferred oxygen levels[17,19,21]. In low oxygen conditions (<10 μM), changes in zooplankton abundance have been linked to oxygen concentration shifts of only a few micromolars[21]. This finely tuned behavior could be disrupted by introducing an oxygenated midwater mining waste plume into the oxyclines or within the OMZ core (see Supplementary Note 1, Table S8 and Fig. S3). A number of zooplankton and micronekton also vertically migrate within the top 1000 m, potentially expanding the zone of a plume's influence[18,19]. Some larger nekton also dive into these depths to forage[42,43], although it should be noted that this may be limited by the strong OMZ in this region. Discharging below the current depth, into the deeper bathypelagic, remains largely unstudied. Limited data suggest that while the bathypelagic is an area of low abundance, it harbors high biodiversity, including species that are distinct from the mesopelagic[44,45]. Furthermore, this community may depend on large particles as the base of its food web, like the processes observed from 700 to 1500 m in this study. Introducing a mining plume into this zone could therefore have comparable adverse effects.

A final potential option is to return the sediment waste to the seafloor. In the case of nodule mining, a seafloor sediment plume will

already be generated by the collector vehicle[46]. A few studies of mining tests suggest that this plume may behave as a turbidity flow, staying within 10's of meters of the seabed[46,47]. Introduction of the mining waste material near the seabed could alter the dynamics of this plume, requiring further study and modelling to evaluate its effects on dispersion. However, this approach may reduce the overall impact footprint by releasing one mining plume instead of two. Given our results showing that a midwater release will disrupt whole food webs, including those linked to seafood sources, bringing the discharge plume back down to the seafloor may minimize environmental risks, and we urge that it be considered.

In conclusion, this study provides empirical evidence that discharge mining waste in midwaters is likely to disrupt midwater food webs. The base of the zooplankton and micronekton food webs at this site, located below the OMZ, rely heavily on large particles (65% of consumer samples). Approximately half of zooplankton taxa in this community are particle feeders, employing suspension feeding or detritivory, and their diets would be severely impacted by the introduction of a midwater mining waste plume composed of similarly-sized particles but of lower nutritional quality. As more than half of the micronekton taxa in this community are zooplanktivores, the micronekton community would also face severe disruption due to the reduction or loss of their zooplankton food source. There are confirmed stress responses to sediment exposure in gelatinous plankton[7], and other risks such as suffocation and toxic metal exposure are also possible. This unique and diverse faunal community is at significant risk if full-scale mining proceeds (Fig. 4). Mitigation is possible through careful selection of discharge depths by mining companies, combined with regulation by national and international agencies.

## Methods

### Ethical guidelines

Vessels employed procedures for collection and combustion of trash as standard to at-sea science research vessels to minimize dumping of trash and disturbance to the ecosystem. Waste disposal at sea followed the internationally recognized standards included in the MARPOL Annex IV and V regulations. All possible efforts were taken to minimize disturbance to the environment.

All animal handling and sampling were state-of-the-art and were in accordance with the American Society of Ichthyologists and Herpetologists "Guidelines for Use of Fishes in Field Research." To minimize pain and stress in the captured animals, they were euthanized as soon as possible after capture if they were still alive. Any that were alive but not needed returned to the ocean as recommended by the ASIH guidelines. The smallest number of animals was used to achieve statistical validity of isotopic analyses for food web analysis. All fishes were collected under UH IACUC permit 14-1934-9. UH IACUC approved this study protocol.

### Study site

Samples were collected in the NORI-D mining claim in the Clarion-Clipperton Zone (CCZ) during 3 cruises: DG5B in Spring (March–April) 2021, DG5C in Fall (October) 2021, and DG7B in Fall (October–November) 2022. In 2021, samples were collected at two sites: the Preservation Reference Zone (PRZ) and the Collector Test Area (CTA). In 2022, sampling was conducted only at the CTA. This study includes particle samples collected during all 3 cruises at both sites, zooplankton samples were collected in Spring and Fall 2021 from the CTA site, and micronekton were collected in Fall 2021 at the CTA.

### Environmental data

Conductivity-temperature-depth (CTD) casts were conducted during all cruises to measure temperature, salinity, transmissometry, and dissolved oxygen concentrations. Oxygen sensors were calibrated using Winkler oxygen titrations on seawater samples collected in Niskin and GoFlo bottles. In situ particle abundance and size distributions were characterized using a LISST (LISST-DEEP, 650 nm, Sequoia Scientific) for 32 logarithmically spaced classes centered between 1.36 and 230.14 µm, with bandwidths ranging from 0.22 to 38 µm. Data were binned to 1db depth intervals and calibrated to the particle minima using the deepest 10 m of the cast.

### Particle collection

Particulate matter was collected using in situ Large Volume Water Transfer Systems (McLane WTS-LV) pumps equipped with tiered filters onto pre-cleaned 0.7 µm glass fiber filters, and 6 µm and 53 µm Nylon or polyester screens to separate particles into 3 size fractions: small (0.7–6 µm), medium (6–53 µm), and large (>53 µm). Filters were subsampled at sea immediately after the sampling cast for bulk and compound-specific isotope analysis, wrapped individually in baked (500 °C, 5 h) aluminum foil, and frozen at −80 °C prior to analysis in the lab (detailed further in Supplementary Note 5).

Background particle samples were collected in the lower mesopelagic (above the proposed discharge depth and below the OMZ core; 800–1000 m) and in the upper bathypelagic zone (1000–1500 m). Thirty one particle samples were collected from below the OMZ core across all 3 cruises; however, only 21 samples (7 small, 7 medium, 7 large) possessed the full suite of amino acids and were used as separate sources in the model. Samples collected when no mining activity was occurring or at locations and depths far from mining activity are referred to as "background samples" in the text. During the Fall 2022 cruise, a small-scale test mining operation was conducted[5], generating a midwater mine waste plume for up to 32 h of sample collection. Pumps were mounted on a rosette equipped with a CTD package (as described above) and a LISST-DEEP to record real-time turbidity and particle size distributions, aiding in plume identification. Tow-Yo casts were performed while the pumps were active to maintain sampling within the plume, approximately 300–1000 m from the point of discharge (detailed further in the Supplementary Notes 5 and 6). These are referred to as "plume samples" in the text. "Discharge samples" were collected aboard the test mining vessel prior to mine waste being discharged at depth. The three same size-fractions of particles were collected.

### Zooplankton collection

Zooplankton were collected using a 1 m² Multiple Opening/Closing Nets and Environmental Sensing System (MOCNESS) from 1500 m to the sea surface. Samples were collected from 9 depth-stratified nets, targeting depths below the OMZ core in the LO and suboxycline (SO) of 1500–1250 m, 1250–1000 m, 1000–800 m, and 800–700 m. Bulk zooplankton was split quantitatively at sea using a Folsom plankton splitter and fractions preserved for isotope and metabarcoding analyses. A ¼ fraction was separated into 5 size fractions, 0.2–0.5 mm, 0.5–1.0 mm, 1.0–2.0 mm, 2.0–5.0 mm, and >5 mm, and stored at −80 °C. These samples were weighed, lyophilized, ground, and stored at −20 °C until stable isotope analysis. In total, 24 zooplankton samples, collected from below the OMZ core in Spring and Fall 2021, were used for isotopic analysis. A ¼-⅛ fraction was preserved in RNAlater, stored at −80 °C, and analyzed by community metabarcoding to assess zooplankton diversity and community composition. A -365-bp fragment of the V1-V2 region of nuclear 18S rRNA was PCR amplified for these analyses, with reads classified using a Naïve-Bayes Classifier in Qiime2[48] using multiple reference databases (detailed further in Supplementary Note 3: DNA Extraction, Amplification, and Sequencing, Bioinformatics; Supplementary Data 3).

### Micronekton collection

Micronekton were collected using a 10 m² MOCNESS equipped with 3 mm mesh nets, sampling from 1500 m to the sea surface. Samples were collected from 5 depth-stratified nets, targeting depths below the

OMZ core in the LO and SO of 1500–1000 m and 1000–700 m. In the field, micronekton were individually sorted to family level, then transported to the lab for further identification to the lowest taxonomic unit possible, often to species level, prior to analysis. Specific micronekton taxa analyzed included *Acanthephyra brevicarinata*, caridean shrimp, *Cyclothone* spp. fish, *Japatella* spp. octopod, and *Eucopia* spp. mysid shrimp. These taxa were selected as abundant representatives collected in both the LO and SO. In total, 21 micronekton samples from below the OMZ core in Fall 2021 were analyzed in this study.

### Stable isotope analysis

Samples for stable isotope analysis were processed using routine procedures[25,49]. Bulk isotope samples were packed into tin capsules and processed using a Costech ECS 2010 Elemental Combustion System coupled to a ThermoFinnigan Delta Plus XP or Thermo Scientific Delta V Advantage via a Thermo Scientific Conflo IV. Reference material (characterized glycine standard and characterized in-house tuna white muscle tissue) was analyzed every 6–12 samples, and the corresponding response factor (ratio of nitrogen peak area, in volt-seconds, to nitrogen reference size, Vs [µg PN]$^{-1}$) of the reference material was used to determine the particulate nitrogen (PN) concentration (µg N-PN) of samples.

For CSIA-AA, samples were hydrolyzed using 6N hydrochloric acid and purified using cation exchange chromatography. Samples were esterified with isopropanol and trifluoracetic acid (TFAA) and then further purified with liquid-liquid extraction using a phosphate buffer: chloroform mixture. Samples were stored in a dichlormethane-TFAA solution at −20 °C and transferred into ethyl acetate before analysis.

Nitrogen isotopic composition was measured using a Thermo Scientific Delta V Plus IRMS interfaced to a Trace gas chromatograph. Internal references L-2-aminoadipic acid (AAA) and L-(+)-norleucine (NOR) with known δ$^{15}$N values were co-injected with each sample, and a reference suite of 14 pure amino acids with known δ$^{15}$N values was co-injected with NOR and AAA every 2–4 sample runs. The corresponding response factors (ratio of AA peak area, in volt-seconds, to AA reference size, Vs [nmol AA]$^{-1}$) of the reference suite were used to determine AA concentration (ng N-AA) of samples. Samples were measured in triplicate when enough material was available.

Total amino acid concentration (as presented in Fig. 1) was calculated by summing the individual amino acid concentrations from CSIA-AA divided by total nitrogen concentration from bulk analysis. A known subsample of the total filter was split for bulk and CSIA-AA, the response factor of the reference material (in either method) was then used to calculate the concentration of the sample analyzed. The subsample percentage (e.g. ½ for bulk analysis, ½ for CSIA-AA) was used to calculate total carbon and nitrogen contents within the sample. Total amino acid content (CSIA-AA, ng N-AA) was normalized to total nitrogen content (Bulk, µg N-PN) in order to compare results between samples.

Carbon isotopic composition of AA was measured using a MAT 253 IRMS interfaced with a Trace GC Ultra. Internal isotopic references AAA, NOR, and an $n$-C$_{20}$ alkane were co-injected with each sample, and a reference suite of 14 pure amino acids with known δ$^{13}$C values was co-injected with NOR and AA every 2-4 samples.

### Data analysis

The suite of AA's used as tracers in this mixing model was selected by iterative statistical tests. First, AA's missing from at least 4 samples per size fraction were excluded. Second, AA's that showed no significant difference between particle size fraction as determined by one-way ANOVA were also excluded. This process resulted in the selection of two nitrogen source amino acids (phenylalanine [Phe] and lysine [Lys]) and one carbon essential amino acid (EAA, leucine [Leu]). Within a particle size fraction, there was no difference between site, cruise, or depth as determined by way of Pairwise PERMANOVA.

The base of the food web was analyzed using a Bayesian isotope mixing model using the package runjags (version 2.2.2-4)[26] in R (version 4.2.2)[50] with dirichletReg (0.7-1)[51]. Three sampling chains were run, each containing 100,000 steps after a 50,000-step adaptation, 40,000-step burn-in, and thinning factor of 50, using an "uninformative"/generalist prior. This mixing model uses δ$^{15}$N$_{Phe}$, δ$^{15}$N$_{Lys}$, and δ$^{13}$C$_{Leu}$ values as tracers. Since these selected amino acids serve as source and essential amino acids, they can be considered non-fractionating during trophic transfer[52,53]. The model assesses the 3 particle size fractions as potential sources that best matched the consumer isotope profile (Source Data 5). Animal samples were run through the model individually by size fraction/taxa (Supplementary Data 4). Given that the amino acid concentrations of the medium and large size fraction plume samples were distinct from the background material, these size fractions were pooled post-modeling to evaluate their mean contribution to the base of the food web (Tables S1 and S2).

Statistical analyses were conducted in R (version 4.2.2)[50], using MASS (version 7.3-58.1)[54], pairwiseAdonis (version 0.4.1)[55], and vegan (version 2.6-4)[56]. Plots were generated with gglopt2 (version 3.5.1)[57].

### Reporting summary

Further information on research design is available in the Nature Portfolio Reporting Summary linked to this article.

### Data availability

All isotopic, Bayesian mixing model, and LISST data generated in this study are publicly available via GitHub and linked to Zenodo (https://github.com/mdowd3/Mining-FoodWeb-CSIA)[58]. Zooplankton sequence data are available in the NCBI Sequence Read Archive under BioProject PRJNA1254332 and SRA accession numbers SRR33375556 to SRR33375942. Isotopic, Bayesian mixing model, and LISST source data generated in this study are provided in Source Data 1–7. Zooplankton and micronekton functional ecology data generated in this study are provided in Supplementary Data 1–2. Zooplankton sequence data generated in this study are provided in Supplementary Data 3. Bayesian consumer data generated in this study are provided in Supplementary Data 4. Source data are provided with this paper.

### Code availability

All model code and dependencies are publicly available via GitHub and linked to Zenodo (https://github.com/mdowd3/Mining-FoodWeb-CSIA)[58]. Bayesian mixing model code (including a readme instruction file, input data, and expected output data) generated in this study is provided in Supplementary Code 1.

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

## Acknowledgements
We thank Natalie Wallsgrove for assistance in isotope analyses and all cruise science teams for assistance at sea. Amanda Merrit designed and illustrated Fig. 4. All authors received support from The Metals Company Inc. (TMC) through its subsidiary Nauru Ocean Resources Inc. (NORI). NORI holds exploration rights to the NORI-D contract area in the CCZ, regulated by the International Seabed Authority and sponsored by the government of Nauru. This is contribution TMC/NORI/D/022 and School of Ocean and Earth Science and Technology contribution number 11,945. All fish were collected under UH IACUC permit 14-1934-9.

## Author contributions
J.C.D., E.G., A.E.W., and B.N.P designed research. M.H.D., V.E.A., A.E.C., J.C.D., E.G., A.E.W., and B.N.P performed research and analyzed data. M.H.D. wrote paper with V.E.A., A.E.C., J.C.D., E.G., A.E.W., and B.N.P.

## Competing interests
The authors declare no competing interests.
