## [Transparent Peer Review file · Nature Communications]

Deep-sea mining discharge can disrupt midwater food webs

Corresponding Author: Mr Michael Dowd

Version 0:

Reviewer comments:

Reviewer #1

(Remarks to the Author)

The goal of this study was to characterize the trophic sources of the faunal food web at the CCZ pelagic community at the lower mesopelagic/upper bathyal depth and to understand if a mining plume will affect the nutritional quality of the trophic sources and if the food web could be disrupted. These goals will help to understand the mining plume impact.

The results are interesting showing that several functional groups will be affected due to a decrease of nutritional quality of the particles, mainly due to dilution with the mining plume particles.

This work is very relevant and timely since it shows clearly evidence of the impact of the mining plume on an ecosystem compartment that will affect upper and lower bathymetric layers since the present organisms are food source for pelagic and bathyal organisms, and some of them are even the connecting link between ecosystems due their daily migrations.

The paper is well written, the methods are appropriated, results are sounding and the authors present good discussion.

Here in some suggestion to clarify and improve the manuscript:

Methods:

-Please specify the type of filters used, how the filters were preserved (some of the information is on the supplementary material, but would help the reader here)

-Also, it would help the reader to specify here how did you measure the amino acids concentrations. You present results as N/PN, but you do not explain what these units are. Is amino acid concentration measure as ratio between Nitrogen and Particulate Nitrogen?

- Bayesian model, you should specify the trophic enrichment factors. You measure bulk stable isotope and Compound specific. Which did you use for the Bayesian model? It is not clear if the bulk or the Compound specific. Maybe a table with the values of source and TEF would be helpful for the reader to reproduce the model.

Results and Discussion:

In the supplementary material you explain the analyses of the functional groups, which are key for the mixing model analyses. Somehow you should include the results and discussion of the functional groups in the main text.

Figure 3- In the caption explain that the graph is the results of the mixing model

Line 171- How did you measure the organic? Which organic are you referring? please explain and rephrase.

line 332- Please rephrase for clearness the sentence Reference material was analysed every 10 samples, and the corresponding response factor of the reference material was used to determine the particulate nitrogen (PN) concentration (g N-PN). What do you mean with response factor? Please explain

Supplementary material

S8- please define SD on the table caption

S10, please explain why just these taxa were used.

Reviewer #2

(Remarks to the Author)

The authors evaluated the potential impact of deep sea mining discharge on midwater food webs via comparison of natural

& mining-associated particles in the context of zooplankton & micronekton communities. The paper is well written, thorough, and highly readable. This study clearly has important implications for future deep sea mining activities and mitigation of their ecological impacts. I have some minor concerns and suggestions for improvement of this manuscript, details below.

Abstract:

Line 29: suggest change to "...community including zooplankton and micronekton" for clarity. Initial wording suggests there aren't any other fauna in the midwater. I anticipate this abstract will be read by non-biologists interested in mining impacts who may not be familiar with mesopelagic communities.

Line 37: "entire community" – as with previous comment, it would be beneficial to (briefly) outline the members of this community. I.e., that the community extends beyond zooplankton and micronekton to large marine predators, etc.

Introduction:

Line 65-66: it would be helpful to give an approximate size range of the particles & fragmented nodules at this point in the paper for reader visualization.

Line 71-74: I don't believe there's a definition of "micronekton" in the ms. Suggest including the size class definition here to increase ms readability across disciplines

Results & Discussion:

Line 119: why are plume and discharge particles combined here?

Line 124: This is the only mention of "phytodetritus" in the paper. This is a useful description of the medium/large particles. Suggest either incorporating this description throughout or remove the single mention.

Line 154 paragraph and associated methods: Regarding the Bayesian mixing model, please expand on why the only source inputs are the 3 particle size fractions collected at depth & the implications of only using those source inputs. Many mesopelagic micronekton undergo DVM to surface waters, where they would have access to alternate basal food web sources. I understand that using only the particles as inputs makes sense in the context of this study's sample collection and am not suggesting a change to the methods. However, I would like to see the potential inputs of other basal sources (e.g., epipelagic phytoplankton) to this food web touched on in the discussion.

Related comment: throughout the ms, the only mention of DVM is in Figure 4's caption. This is an important aspect of the community to touch on at least briefly in the context of mesopelagic food web dynamics & basal sources.

Line 168: repeated use of "large" is a bit awkward for readability.

Line 171: please expand on why the concentration of AAs matters biologically to increase ms readability across disciplines

Line 205-206: suggest removing the ## values inside the parenthetical – they are nearly identical to the primary values and it is hard to read so many numbers in one (long) sentence. Leave the parenthetical ##s to the supplemental

Line 213: horizontal or vertical migration away from the impacted area?

Line 227: note that many top predators travel to the sampled mesopelagic/bathypelagic depths as well

Line 253-261: suggest reducing the number of numeric values in this concluding paragraph for readability as they are distracting from the larger themes. E.g., remove the "(85%...)" parenthetical in line 260

Line 262-263: need more than one citation in this sentence if it refers to "other studies" plural

Line 266: suggest change "and/or" to "alongside". Surely both careful decision making by mining companies AND some level of regulation by agencies are needed

Line 288 (and 97): it would be helpful to have the background particle collection depths here (without needing to go searching in the SI), as they are important to interpretation of the study results.

Figure 1: This may be a problem with my computer, but this figure appears slightly blurry. Suggest increasing text size in the legend box

Figure 2: suggest moving legend into white space of panel B for space efficiency, especially as the figure text is very small

Figure 4: the smiley faces on the copepods are charming but perhaps distracting/emotionally biasing from the main purpose of the figure.

Line 559: "vertical migration that provisions commercially harvested fishes" text is confusing. Which fisheries are you referring to? Suggest ending the sentence after "migration"

Overall a well-designed study with interesting and important results.

Version 1:

Reviewer comments:

Reviewer #1

(Remarks to the Author)

I am happy with the way the authors improved the manuscript with my suggestions and reviewer #2.

The paper is well-written, comprehensive, and highly accessible. It offers valuable insights with clear implications for future deep-sea mining operations and the mitigation of their ecological impacts.

I recommend the publication of the manuscript NGS-2013-03-00534.

(Remarks on code availability)

Reviewer #2

(Remarks to the Author)

The authors have nicely revised the manuscript. I recommend it for publication

(Remarks on code availability)

The code is well organized & commented with a clear README file. I was able to install and run the code

Reviewers' comments and **response to reviewers (in red)** for manuscript NGS-2013-03-00534.

Reviewer 1

The goal of this study was to characterize the trophic sources of the faunal food web at the CCZ pelagic community at the lower mesopelagic/upper bathyal depth and to understand if a mining plume will affect the nutritional quality of the trophic sources and if the food web could be disrupted. These goals will help to understand the mining plume impact. The results are interesting showing that several functional groups will be affected due to a decrease of nutritional quality of the particles, mainly due to dilution with the mining plume particles.

This work is very relevant and timely since it shows clearly evidence of the impact of the mining plume on an ecosystem compartment that will affect upper and lower bathymetric layers since the present organisms are food source for pelagic and bathyal organisms, and some of them are even the connecting link between ecosystems due their daily migrations.

The paper is well written, the methods are appropriated, results are sounding and the authors present good discussion.

We thank the reviewer for their kind words and for her/his careful reading of our manuscript and providing useful comments and critiques.

Here in some suggestion to clarify and improve the manuscript:

Methods:

-Please specify the type of filters used, how the filters were preserved (some of the information is on the supplementary material, but would help the reader here)

We used 0.7 μm glass fiber filters, 6 and 53 μm Nylon and polyester screens. The Nylon and polyester screens were cleaned using sequential rinses with HCl, DI and MeOH, while the glass fiber filters were combusted at 500°C for 5 hours. Filters were subsampled at sea immediately after the sampling cast for bulk and compound-specific analysis, wrapped individually in baked (500°C 5 hours) aluminum foil, and frozen at -80°C prior to analysis in the lab. These details have been added to the "Particle Collection" section of the Methods (line 376-381). We also clarified in this section that the full methodology is described in the Supplementary Materials.

-Also, it would help the reader to specify here how did you measure the amino acids concentrations. You present results as N/PN, but you do not explain what these units are. Is amino acid concentration measure as ratio between Nitrogen and Particulate Nitrogen?

We added a paragraph in the "Stable Isotope Analysis" section, positioned after the CSIA-AA Nitrogen protocol and before the CSIA-AA carbon protocol, to explain how we measured the concentration of amino acids (line 458-466). Briefly, we calculated the concentration of each amino acid from the combined peak area for all masses. The peak area was calibrated using reference compounds (in nanograms). The total AA content from in a sample was divided by the total N content (in micrograms) from bulk analysis in the same sample. This ratio is used to facilitate comparisons between samples.

- Bayesian model, you should specify the trophic enrichment factors. You measure bulk stable isotope and Compound specific. Which did you use for the Bayesian model? It is not clear if the bulk or the Compound specific. Maybe a table with the values of source and TEF would be helpful for the reader to reproduce the model.

The input to the Bayesian mixing model were $\delta^{15}\text{N}_{\text{Phe}}$, $\delta^{15}\text{N}_{\text{Lys}}$, and $\delta^{13}\text{C}_{\text{Leu}}$, values all collected through CSIA-AA; no bulk isotope data were used. We selected these amino acids because they are source and essential amino acids, which have been shown in the literature to be non-fractionating during trophic transfers. Consequently, no trophic discrimination factors need be applied. We added text in the paragraph discussing the mixing model parameters to clarify this (line 485-488).

Results and Discussion:

In the supplementary material you explain the analyses of the functional groups, which are key for the mixing model analyses. Somehow you should include the results and discussion of the functional groups in the main text.

We respectfully disagree with the reviewer's suggestion. Including additional detailed results and discussion of the functional groups in the main text would unnecessarily lengthen the manuscript and divert attention from its primary focus. Furthermore, separate manuscripts that specifically address the ecology of zooplankton and micronekton are currently in preparation and will be submitted to more specialized journals.

Figure 3- In the caption explain that the graph is the results of the mixing model

We added text that this plot shows the mixing model mean contribution of particles in the figure title and caption.

Line 171- How did you measure the organic? Which organic are you referring? please explain and rephrase.

We apologize for the confusion. This refers to the total AA content in the particles and should have stated "significantly lower in amino acid content". We have removed the word "organic" for clarity and added the relevant figure number to those sentences (line 211-215). The following sentence, which essentially repeated the information about higher particle concentrations within the plume compared to the background, has been replaced with content aimed at improving readability for reader from other disciplines, as requested by Reviewer #2.

line 332- Please rephrase for clearness the sentence Reference material was analyzed every 10 samples, and the corresponding response factor of the reference material was used to determine the particulate nitrogen (PN) concentration ($\mu\text{g N-PN}$). What do you mean with response factor? Please explain

The response factor is based on the peak area (measured in volt-seconds, Vs) and known concentration of the reference materials, and is used to determine the concentration of individual amino acids in the sample. The same procedure is used to determine the concentration of particulate carbon and nitrogen content using Elemental Analysis-Isotope ratio mass spectrometry. We added text to clarify this, including the appropriate units: (Vs [$\mu\text{g PN}$]⁻¹) for bulk particulate nitrogen (line 438-439) and (Vs [nmol AA]⁻¹) for CSIA-AA (line 451-452).

Supplementary material

S8- please define SD on the table caption

We added a sentence to the Table S8 caption clarifying SD is the standard deviation of the grouped results.

S10, please explain why just these taxa were used.

Taxa were selected as they were abundant representatives of different functional groups collected in both the Lower Oxycline and Suboxycline, as described in the Methods Section. We added a sentence to Table S10 caption clarifying this.

Reviewer 2

The authors evaluated the potential impact of deep sea mining discharge on midwater food webs via comparison of natural & mining-associated particles in the context of zooplankton & micronekton communities. The paper is well written, thorough, and highly readable. This study clearly has important implications for future deep sea mining activities and mitigation of their ecological impacts. I have some minor concerns and suggestions for improvement of this manuscript, details below.

We thank the reviewer for their thoughtful comments. We have re-written parts of the manuscript to make it more understandable to a general audience. Responses to specific comments follow.

Abstract:

Line 29: suggest change to "...community including zooplankton and micronekton" for clarity. Initial wording suggests there aren't any other fauna in the midwater. I anticipate this abstract will be read by non-biologists interested in mining impacts who may not be familiar with mesopelagic communities.

Excellent suggestion. We added "including" as suggested (line 31).

Line 37: "entire community" – as with previous comment, it would be beneficial to (briefly) outline the members of this community. I.e., that the community extends beyond zooplankton and micronekton to large marine predators, etc.

We added "extending beyond zooplankton and micronekton to nekton, including large marine predators" (line 37-39).

Introduction:

Line 65-66: it would be helpful to give an approximate size range of the particles & fragmented nodules at this point in the paper for reader visualization.

We added the typical size range of <2 – 63 μm to this sentence, and include an appropriate citation (line 78).

Line 71-74: I don't believe there's a definition of "micronekton" in the ms. Suggest including the size class definition here to increase ms readability across disciplines

We added the clarification "small fish, cephalopods, and crustacea, ~2-20 cm in size" in the last sentence, with citation, to increase readability (line 96).

Results & Discussion:

Line 119: why are plume and discharge particles combined here?

We were allocated limited time on the mining test cruise to collect plume and discharge particle samples. Of those particles collected, we found no difference between plume and discharge particles. Therefore, these sample types were combined, providing a direct comparison between

background particle samples and mining-associated particles. We changed “plume particles” in the first sentence of this paragraph to “mining-associated particles” to avoid ambiguity (line 148).

Line 124: This is the only mention of “phytodetritus” in the paper. This is a useful description of the medium/large particles. Suggest either incorporating this description throughout or remove the single mention.

Thank you for this observation. We removed the term “phytodetritus” since medium/large particles could also be composed of fecal material or aggregations, and this paper does not explicitly explore the origins and composition of the particles, although this will be a subject of a subsequent manuscript.

Line 154 paragraph and associated methods: Regarding the Bayesian mixing model, please expand on why the only source inputs are the 3 particle size fractions collected at depth & the implications of only using those source inputs. Many mesopelagic micronekton undergo DVM to surface waters, where they would have access to alternate basal food web sources. I understand that using only the particles as inputs makes sense in the context of this study’s sample collection and am not suggesting a change to the methods. However, I would like to see the potential inputs of other basal sources (e.g., epipelagic phytoplankton) to this food web touched on in the discussion.

We agree with this comment and have added text to the manuscript addressing this point. Acknowledging the presence of DVM, we had already considered its potential influence and took steps to minimize its impact. As described in the revised text, all samples were selected from nighttime tows to preferentially target non-migratory resident species. While this manuscript only briefly mentions it, ongoing studies (manuscripts in preparation) will explore this topic in greater detail with additional data. Current evidence indicates that only ~20% of the zooplankton biomass present below the OMZ, in the deep mesopelagic, migrates to the upper ocean at night. Lastly, the micronekton taxa analyzed were specifically selected for their non-migratory behavior, further limiting potential contributions from epipelagic or DVM-influenced trophic sources (line 193-200).

Related comment: throughout the ms, the only mention of DVM is in Figure 4’s caption. This is an important aspect of the community to touch on at least briefly in the context of mesopelagic food web dynamics & basal sources.

We agree with this suggestion and have now added text to the Introduction (lines 72-74) with appropriate citations that acknowledges that DVM vertically mixes food webs. We have also added clarity at lines 193-200 regarding the rationale for a limited influence of DVM on our meso- and bathypelagic material. Lastly, we have added text (lines 298-300) that describes how DVM may expand the zone of influence of the mine waste plume.

Line 168: repeated use of “large” is a bit awkward for readability.

Thank you. We changed this sentence to: “In sum, our mixing model indicates that large particles make a substantial contribution to the base of the food web below the OMZ (Fig 3), a region expected to be impacted by mining waste plumes.” (Line 211-213)

Line 171: please expand on why the concentration of AAs matters biologically to increase ms readability across disciplines

We added a sentence explaining that reduced AA concentration is a key indicator of the reduced nutritional value of particles, leading to less protein intake for the same foraging effort. (Line 215-218).

Line 205-206: suggest removing the ## values inside the parenthetical – they are nearly identical to the primary values and it is hard to read so many numbers in one (long) sentence. Leave the parenthetical ##s to the supplemental

We removed ##'s inside parenthetical as suggested to improve readability. They are also present in supplemental materials.

Line 213: horizontal or vertical migration away from the impacted area?

This is true for the more mobile micronekton, although we suspect that less mobile animals may not be able to easily swim away from the plume, and we now say this on line 270. Zooplankton would only be capable of vertical repositioning relative to the plume.

Line 227: note that many top predators travel to the sampled mesopelagic/bathypelagic depths as well

This is a good point, although the depths of migration for a number of taxa are likely shallower in this region due to the presence of the pronounced OMZ. There isn't space to go into the details, but we agree with the reviewer on the general point and have added a statement and citations on line 300-302.

Line 253-261: suggest reducing the number of numeric values in this concluding paragraph for readability as they are distracting from the larger themes. E.g., remove the “(85%...)” parenthetical in line 260

Thank you for the suggestion. We removed (85%), changed 53% of zooplankton taxa to “Approximately half” and 60% of micronekton taxa to “more than half”. We kept 65% of

consumer samples as this is based on the isotopic mixing model and is the main focus of the paper. (line 325-329)

Line 262-263: need more than one citation in this sentence if it refers to “other studies” plural

We changed “other studies have also confirmed...” to “there are confirmed...”. (line 331)

Line 266: suggest change “and/or” to “alongside”. Surely both careful decision making by mining companies AND some level of regulation by agencies are needed

We changed this sentence to read: “Mitigation is possible through careful selection of discharge depths by mining companies, combined with regulation by national and international agencies.” (line 334-336).

Line 288 (and 97): it would be helpful to have the background particle collection depths here (without needing to go searching in the SI), as they are important to interpretation of the study results.

We added depth ranges of 800-1500m on line 127-128, and clarifying lower mesopelagic below the OMZ core 800-1000m and upper bathypelagic 1000-1500m on line 382-384.

Figure 1: This may be a problem with my computer, but this figure appears slightly blurry. Suggest increasing text size in the legend box

We changed the figure so that the legend text size is increased to be the same size as the axis title and particle size label. We slightly increased the width of the figure to accommodate the increase in legend text size.

Figure 2: suggest moving legend into white space of panel B for space efficiency, especially as the figure text is very small

We moved the legend as recommended, and increased the text size to improve readability.

Figure 4: the smiley faces on the copepods are charming but perhaps distracting/emotionally biasing from the main purpose of the figure.

We aim to convey that the health of zooplankton affected by the mining plume may be compromised. We regretfully agree with Reviewer #2 that, while the smile and frown on the copepods are charming, they may be distracting and introduce emotional bias. Live copepods hold their antennules perpendicular to their body; however, after death, the antennules typically droop and lie more parallel to the body—a characteristic “dead pose.” We have included this visual cue to indicate “unhappy” copepods in the plume-influenced food web, but recognize that it may be too subtle for those unfamiliar with zooplankton. We note that X-shaped eyes are a

widely recognized cartoon convention for depicting dead or dying animals. Therefore, we use both droopy antennules and X-eyes to convey compromised zooplankton health in this figure.

Line 559: “vertical migration that provisions commercially harvested fishes” text is confusing. Which fisheries are you referring to? Suggest ending the sentence after “migration”

We ended sentence after “migration” as suggested.

Overall a well-designed study with interesting and important results.